# Knockdown of the Sodium/Potassium ATPase Subunit Beta 2 Reduces Egg Production in the Dengue Vector, *Aedes aegypti*

**DOI:** 10.3390/insects14010050

**Published:** 2023-01-05

**Authors:** Nathan P. Martinez, Matthew Pinch, Yashoda Kandel, Immo A. Hansen

**Affiliations:** Department of Biology, New Mexico State University, Las Cruces, NM 88003, USA

**Keywords:** NKAβ2, sodium/potassium ATPase, cationic amino acid transporter, *Aedes aegypti*, nutrient sensor, fecundity

## Abstract

**Simple Summary:**

The yellow fever mosquito, *Aedes aegypti*, is a vector for several viruses including yellow fever, Zika, and dengue virus, which infect millions of people worldwide every year. Novel methods for population control of mosquitoes are an important research topic. The sodium/potassium ATPase pump is vital for ion transport across cell membranes, making it indispensable to many biological processes including nervous signaling and muscle contraction. This pump contains three subunits: alpha, beta, and FXYD, with the alpha subunit serving to transport ions, and the beta and FXYD subunits serving to regulate the function of the alpha subunit. In this study, we analyzed tissue expression of the sodium/potassium ATPase beta 2 subunit in *Ae. aegypti.* We also knocked down expression of this gene and measured the fecundity and mortality of treated mosquitoes. Expression levels of sodium/potassium ATPase beta 2 were highest in ovaries compared to other analyzed tissues. The total number of eggs laid by sodium/potassium ATPase beta 2 knockdown mosquitoes were significantly less than the control treated group. Overall, our findings suggest that sodium/potassium ATPase beta subunit 2 plays a role in regulating egg development in *Ae. aegypti*, and may therefore serve as a target for future mosquito control studies.

**Abstract:**

The Na^+^/K^+^ ATPase (NKA) is present in the cellular membrane of most eukaryotic cells. It utilizes energy released by ATP hydrolysis to pump sodium ions out of the cell and potassium ions into the cell, which establishes and controls ion gradients. Functional NKA pumps consist of three subunits, alpha, beta, and FXYD. The alpha subunit serves as the catalytic subunit while the beta and FXYD subunits regulate the proper folding and localization, and ion affinity of the alpha subunit, respectively. Here we demonstrate that knockdown of NKA beta subunit 2 mRNA (*nkaβ2)* reduces fecundity in female *Ae. aegypti*. We determined the expression pattern of *nkaβ2* in several adult mosquito organs using qRT-PCR. We performed RNAi-mediated knockdown of *nkaβ2* and assayed for lethality, and effects on female fecundity. Tissue expression levels of *nkaβ2* mRNA were highest in the ovaries with the fat body, midgut and thorax having similar expression levels, while Malpighian tubules had significantly lower expression. Survival curves recorded post dsRNA injection showed a non-significant decrease in survival of *nkaβ2* dsRNA-injected mosquitoes compared to GFP dsRNA-injected mosquitoes. We observed a significant reduction in the number of eggs laid by *nkaβ2* dsRNA-injected mosquitoes compared to control mosquitoes. These results, coupled with the tissue expression profile of *nkaβ2,* indicate that this subunit plays a role in normal female *Ae. aegypti* fecundity. Additional research needs to be conducted to determine the exact role played by NKAβ2 in mosquito post-blood meal nutrient sensing, transport, yolk precursor protein (YPP) synthesis and yolk deposition.

## 1. Introduction

The Na^+^/K^+^-ATPase (NKA) is a member of the P-Type family of ATPase transport proteins [1]. The predominant function of NKA is active transport using ATP hydrolysis to facilitate the efflux of sodium ions and influx of potassium ions in eukaryotic cells. The establishment of sodium and potassium ion gradients across the plasma membrane by NKA is necessary for several homeostatic processes. These ion gradients can be used by cells to power secondary active transport, using the energy of the flow of either sodium or potassium ions back down their concentration gradients to power the transport of other substrates against their concentration gradients [2]. Recovery of the resting membrane potential in excitatory cells after action potential propagation is another important function of NKA pumps, making them indispensable for coordinated neural signaling and muscle contraction [3]. As a central transporter involved in maintaining the electrochemical gradient of multiple ions, NKA pumps also play a vital role in maintenance of cellular osmolarity, and by extension, cell volume [4,5]. In addition to ion transport functions, NKA pumps play a role in cell–cell adhesion through association of N-linked glycans on the extracellular domains of their beta subunits. [6,7]. Finally, populations of NKA pumps localized to caveolae in cell membranes can act as receptors for a class of molecules called cardiac glycosides, demonstrating their importance in cell signaling in addition to their vital role in the maintenance of electrochemical gradients [8,9]. 

NKA is comprised of three independent subunits: alpha, beta, and FXYD. The alpha subunit is the catalytic subunit, containing 10 transmembrane domains [10], and it utilizes energy from ATP hydrolysis to pump sodium ions out of the cell and potassium ions into the cell at a rate of 3:2 [11,12]. Mammalian genomes contain multiple alpha subunit genes which code for tissue-specific subunit expression. The alpha one isoform is largely distributed within all cell types [13,14], the alpha 2 isoform is predominantly located in the muscle, heart, and brain [13,14,15,16,17], the alpha 3 isoform is predominantly found within nervous tissue [13,16,18], and the alpha 4 isoform is predominately located in the spermatozoa [19]. In contrast to vertebrates, many insects only possess a single gene coding for the alpha subunit. However, this single gene codes for many splice variants, allowing for potentially similar levels of complexity in expression and function. For example, the *Drosophila melanogaster* NKA alpha gene (Entrez ID: 48971) has 11 annotated splice variants, while the *Aedes aegypti* NKA alpha gene (Entrez ID: 5575790) has over 20 annotated splice variants. 

While the remaining two subunits in the NKA protein do not play a direct role in ion transport, they are important for regulating the proper folding, localization, and function of the alpha subunit. Additionally, multiple genes code for different isoforms of each regulatory subunit, allowing for complex assemblies of different subunit isoforms in tissue-specific or function-specific patterns. The beta subunit is composed of one transmembrane domain with a large glycosylated extracellular region that facilitates maturation and translocation of the catalytic alpha subunit to the plasma membrane [20,21,22] and is also involved in establishing tight junctions between cells through interaction with other beta subunits [23]. All animals possess multiple genes coding for unique beta subunit isoforms. The beta 1 isoform may play a role in oxidative stress [24], while the beta 2 isoform is involved with cell–cell contact and dominantly facilitates the establishment of the electrochemical gradient by reducing the potassium affinity and raising the binding affinity to extracellular sodium [25]. The final subunit of the oligomeric protein is the FXYD subunit which, like the beta subunit, has many isoforms coded for by different genes, and regulates the function of NKA by affecting the affinity for potassium ions [26,27]. 

The fat body of the yellow fever mosquito, *Ae. aegypti* plays an important role in metabolism by absorbing nutrients from digested blood meals to incorporate them into newly synthesized macromolecules such as proteins, lipids, and carbohydrates [28]. These nutrients can be stored for later use, but most of these molecules are exported from the fat body to be deposited as yolk during vitellogenesis, or the development of a clutch of eggs. As the fat body plays a central role in vitellogenesis, this organ must maintain a robust nutrient sensor system to monitor for changes in organismal nutrient content and respond accordingly. To this end, fat body cells integrate input from several signaling pathways including the hormone signals, insulin-like hormone, juvenile hormone and 20-hydroxyecdysone, as well as intracellular signaling mediated by the mechanistic target of rapamycin (mTOR) network (reviewed in [29]). The main role of nutrient sensors is to detect extracellular and intracellular nutrient availability and induce cellular signal transduction leading to changes in cellular function [30,31]. In the *Ae. aegypti* fat body, a major nutrient sensor is the mTOR complex 1 (mTORC1), which has been demonstrated to play an indispensable role in yolk precursor protein (YPP) synthesis during vitellogenesis [29,32,33]. In response to hormone signaling, and increased nutrient influx after blood feeding, the TOR kinase phosphorylates the p70-S6 Kinase, which in turn phosphorylates ribosomal protein S6, stimulating translation of YPPs [29,32,33]. 

Amino acid influx through amino acid transporters, particularly the family of cationic amino acid transporters (CATs) is necessary for mTORC1 activation and initiation of YPP synthesis [34]. The CAT family in *Ae. aegypti* consists of five genes coding for transceptors that sense extracellular cationic amino acids and transport them into the cell where they can be used for new protein synthesis [35,36]. Previous research has demonstrated the importance of CATs in vitellogenesis [37] and has identified a set of putative protein interactors for one of the CAT family members, CAT1 [38]. These potential interaction partners include NKA subunit beta 2 (Entrez ID: 5573862; NKAβ2) [38]. As NKA has been demonstrated to play a variety of important roles, including regulation of cellular tonicity, maintenance of charge balance across the plasma membrane and signal reception, this putative interaction between NKAβ2 and CAT1 is of interest to determine if NKA pumps play a role in *Ae. aegypti* vitellogenesis.

In this study, we determined the expression pattern of NKAβ2 in multiple structures of adult female *Ae. aegypti* using quantitative RT-PCR (qRT-PCR). We then used RNAi to knockdown *nkaβ2* expression in adult female mosquitoes and assayed for changes in fecundity and flight performance in NKAβ2 knockdowns relative to controls. Our results show that NKAβ2 has tissue-specific expression in female *Ae. aegypti*, and loss of NKAβ2 in adult females reduces clutch sizes after a blood meal.

## 2. Materials and Methods

### 2.1. Mosquito Rearing

*Aedes aegypti* Liverpool mosquitoes were reared under standard laboratory conditions as previously described [38]. Briefly, eggs were hatched in pans of deionized water, and larvae were kept at 27 °C and fed on dry cat food pellets. Pupae were collected and stored in Bug Dorm-1 insect cages (Bugdorm, Taichung, Taiwan). After emergence, adults were allowed to feed on 20% sucrose solution ad libitum. All adults were maintained under standard conditions (14 h: 10 h light-dark cycle, 27 °C, 80% humidity) in our insectary. All experiments were performed on mosquitoes five to seven days post-eclosion. For blood feeding experiments, female mosquitoes were fed on defibrinated bovine blood (HemoStat Laboratories, Dixon, CA, USA) warmed to 37 °C for 1 h.

### 2.2. Total RNA Isolation and cDNA Synthesis

Total RNA was isolated from either three (*n* = 3) pools of 10 organs and structures (determination of *nkaβ2* expression profile), or three (*n* = 3) pools of 10 mosquitoes with heads removed (determination of *nkaβ2* dsRNA knockdown effect) using a Qiagen RNeasy kit (Qiagen, Germantown, MD, USA) following the manufacturer’s protocol. Total RNA concentrations were measured using a NanoDrop 1000 (Thermo Scientific, Waltham, MA, USA). To generate cDNA, 250 ng of RNA from each sample was reverse-transcribed using Bio-Rad iScript^TM^ Reverse Transcription Supermix for RT-qPCR (Bio-Rad, Hercules, CA, USA) following the manufacturer’s instructions. To control for potential genomic DNA contamination, a set of non-reverse transcribed (noRT) samples was generated using iScript^TM^ noRT Supermix (Bio-Rad, Hercules, CA, USA). 

### 2.3. qRT-PCR

Primers for *nkaβ2* (Gene ID: 5573862) and *ribosomal protein s7* (*rps7*) were designed using PrimerBLAST [39] and NetPrimer (Premier Biosoft, Palo Alto, CA, USA) as previously described [40] (see Table 1 for primer sequences). 

All qRT-PCR reactions were prepared as previously described [38,40]. Briefly, cDNA samples were diluted 1:1 in nuclease-free water, and mixed with iTaq Universal SYBRgreen Supermix (Bio-Rad, Hercules, CA, USA), qRT-PCR primers, and nuclease-free water following the manufacturer’s instructions. Reactions were prepared with two technical replicates for each sample, and one noRT reaction for each sample to control for genomic DNA contamination. Each plate was run on a CFX96 Touch Real-Time PCR Detection System (Bio-Rad, Hercules, CA, USA) with the following protocol: initial denaturation at 95 °C for 30 s followed by 40 cycles of denaturation at 95 °C for 5 s and combined annealing/elongation at 60 °C for 30 s. SYBRgreen fluorescence was detected after each elongation step. Immediately after each PCR completed, melting curves were run on each plate with the following parameters: 65–95 °C temperature ramp in 0.5 °C increments with a five second hold at each step. Raw fluorescence data from each qRT-PCR run were analyzed using Bio-Rad CFX Maestro software (Bio-Rad, Hercules, CA, USA). Average C_q_ values from technical replicates of each sample were analyzed by normalization of *nkaβ2* data to *rps7* data prior to quantification of changes in *nkaβ2* expression. Relative gene expression was determined using the 2^−ΔΔCT^ method [41] using the average ovary ΔC_T_ as a calibrator for tissue expression samples, and average GFP ΔC_T_ as a calibrator for dsRNA-injected samples.

### 2.4. Na^+^/K^+^ ATPase Subunit β2 (nkaβ2) Annotation

Phylogenetic analysis of *Ae. aegypti* NKA alpha and beta subunit protein sequences was performed as previously described [38]. *Ae. aegypti* NKA alpha and beta subunit protein sequences were accessed NCBI protein database. *Drosophila melanogaster* NKA alpha subunit and Nervana (beta subunits) protein sequences were also accessed from the NCBI protein database. All protein sequences were aligned using the MUSCLE algorithm in MEGA version 11 [42], and a neighbor-joining tree was constructed to visualize the orthology between *Ae. aegypti* and *D. melanogaster* NKA subunits.

### 2.5. RNAi Knockdown of nkaβ2

Double-stranded RNA (dsRNA) targeting *nkaβ2* and targeting GFP were synthesized to perform RNAi knockdowns [43]. Anti-GFP dsRNA was used as a negative injection control since *Ae. aegypti* do not have a gene encoding GFP, and therefore, anti-GFP dsRNA should not target native mosquito mRNAs. Briefly, dsRNA synthesis primers for *nkaβ2* were designed using PrimerBLAST, and T7 promoter sequences were added on to the 5′ ends of both the forward and reverse primers to facilitate in vitro dsRNA synthesis. GFP dsRNA primers were taken from a previous study [44]. All dsRNA synthesis primers are listed in Table 2. T7-tagged *nkaβ2* DNA was generated by PCR using whole mosquito cDNA as a template, and the PCR-generated fragment was cloned into the TOPO4 expression vector. Cloned plasmids were transformed into chemically competent *E. coli*, plated on LB agar with ampicillin, and incubated overnight at 37 °C. Individual colonies were picked and transferred to 5 mL LB broth with ampicillin and grown overnight at 37 °C and 220 rpm shaking. Plasmids were extracted from bacteria using a Qiagen Plasmid Mini kit (Qiagen, Germantown, MD), and the presence of the *nkaβ2* insert was validated by sequencing (MCLAB, South San Francisco, CA, USA). TOPO plasmids with confirmed *nkaβ2* inserts were used as templates for the synthesis of T7-tagged *nkaβ2* template DNA. A GFP-containing plasmid (3′ EGFP pXOON) was used as the template to produce T7-tagged GFP dsDNA. T7-tagged dsDNA fragments were generated by PCR, and PCR product sizes were confirmed by agarose gel electrophoresis prior to purification using a Qiagen PCR Purification kit (Qiagen, Germantown, MD, USA). Purified products were then used as the template for dsRNA synthesis using a Megascript^TM^ RNAi Kit (ThermoFisher Scientific, Waltham, MA, USA). dsRNA synthesis reactions were allowed to run for four hours, frozen overnight, and finished for another four hours for a total of minimum eight hours of synthesis time. Reaction mixes, nuclease digestion, and dsRNA clean-up were performed following the kit instructions.

Adult female mosquitoes (5–7 days post-eclosion) were injected with dsRNA as previously described [38]. Briefly, mosquitoes were injected intrathoracically using a mouth injector with approximately 1 µg (concentration of 1µg/µL) of either *nkaβ2* or GFP dsRNA and given one hour to recover. Surviving mosquitoes were maintained as described above until use in other experiments. Effectiveness of dsRNA knockdown of *nkaβ2* was assayed using qRT-PCR with total RNA extracted from whole mosquitoes using the qRT-PCR methods described above.

### 2.6. Mortality Assay

To determine the effects of *nkaβ2* knockdown on mosquito mortality, a survival assay was conducted on mosquitoes injected with either *nkaβ2* dsRNA (*n* = 73) or GFP dsRNA (*n* = 67). For a 72 h period post-injection (PI), mosquito mortality was counted every 24 h for both the GFP dsRNA injected mosquitoes and the *nkaβ2* dsRNA injected mosquitoes. After each count was performed, dead mosquitoes were removed to prevent interference with future mortality counts.

### 2.7. Ovary Morphology Assay

Groups of adult female mosquitoes were injected with either *nkaβ2* or GFP dsRNA as described above. Three days PI, mosquitoes from each treatment group were given a blood meal as described in the “Mosquito rearing” section above. Immediately prior to feeding, 10 mosquitoes from each group were collected for use as unfed baseline samples. Ovaries from 10 blood fed mosquitoes were dissected at 24, 48, and 72 h PBM, and both ovaries from each mosquito were imaged using an Olympus SZX12 stereo microscope (Olympus Life Science, Waltham, MA, USA). Ovary lengths were measured in ImageJ [45] using a scale of 1 mm, and ovary pairs from each mosquito were averaged to determine the mean ovary length of each individual mosquito. The mean ovary lengths from all mosquitoes in each treatment/time group were averaged to determine the change in ovary length over time in *nkaβ2* dsRNA-injected and GFP dsRNA-injected mosquitoes. Additionally, five follicles from each mosquito were measured to determine the change in average follicle length over time in *nkaβ2* dsRNA-injected and GFP dsRNA-injected mosquitoes. The follicle lengths from each mosquito were averaged to generate mean follicle lengths for each mosquito, and the mean lengths from all mosquitoes in each treatment/time group point were averaged to determine the change in follicle length over time.

### 2.8. Clutch Size and Hatch Rate Assay

Groups of adult female mosquitoes were injected with either *nkaβ2* or GFP dsRNA as described above. Three days PI, mosquitoes from each treatment group were given a blood meal as described in the “Mosquito rearing” section above. Individual engorged mosquitoes were moved to egg laying chambers and allowed to lay eggs as previously described [38]. Egg papers with eggs were desiccated for 96 h, and then egg numbers were counted using an Olympus SZX12 stereo microscope (Olympus Life Science, Waltham, MA, USA). After counting, egg clutches were desiccated for at least 72 additional hours, and then egg clutches were hatched under the same conditions as described in the “Mosquito rearing” section above. Numbers of larvae from each clutch were counted and compared to the egg count from their clutch to determine hatch rates.

### 2.9. Statistical Analysis

Normality tests of the distributions of all data were performed with Shapiro–Wilk tests prior to selection of additional statistical analyses. A one-way ANOVA was used to test for statistically significant differences in tissue and structure-specific *nkaβ2* localization measured by qRT-PCR. Statistical significance of RNAi knockdown of *nkaβ2* relative to GFP control was determined using an unpaired t-test. Differences in percent survival of *nkaβ2* dsRNA-injected and GFP dsRNA-injected mosquitoes was determined using a Mantel–Cox log-rank test. Statistical significance in average ovary lengths and follicle lengths between *nkaβ2* dsRNA-injected and GFP dsRNA-injected mosquitoes was determined using unpaired t-tests at each time point. Significant differences in clutch sizes and hatch rates between *nkaβ2* dsRNA-injected and GFP dsRNA-injected mosquitoes was determined using Mann–Whitney U tests. For all assays, a p-value threshold of *p* ≤ 0.05 was considered statistically significant. All statistical analyses were performed using GraphPad Prism8 software (GraphPad Software, San Diego, CA, USA).

## 3. Results

### 3.1. nkaβ2 Expression Is High in Ovaries of Female Ae. aegypti

We profiled *nkaβ2* expression in five tissues and structures of female *Ae. aegypti* by qRT-PCR: fat bodies (FB), midguts (MG), Malpighian tubules (MT), ovaries (OV), and thoraxes (TX) (Figure 1a). The various tissue types analyzed showed no significant difference in *nkaβ2* expression levels (Figure 1a). The expression of *nkaβ2* was highest in ovaries relative to the other four tissue types (FB, MG, TX, MT) (Figure 1a).

### 3.2. nkaβ2 Is an Ortholog of Drosophila Melanogaster Nervana 3

The *D. melanogaster nervana* (*nrv*) gene family (*nrv1*, *nrv2*, and *nrv3*) are orthologs of the *Ae. aegypti nka beta subunit* genes, so we predicted that our *nkaβ2* gene may serve a similar function as its *nrv* ortholog. Therefore, we performed phylogenetic analysis of all *Ae. aegypti* NKA alpha and beta subunit genes annotated in the NCBI Gene database using the *D. melanogaster* NKA alpha subunit and *nrv* genes and determined that our *nkaβ2* (Gene ID: 5573862) gene is orthologous *nrv3* (Appendix A).

### 3.3. dsRNA Reduces nkaβ2 Transcript Levels in Mosquitoes

We injected groups of female *Ae. aegypti* mosquitoes with approximately 1 µL of either anti-GFP dsRNA or anti-*nkaβ2* dsRNA at a concentration of 1 µg/µL. Three days post-injection, we sacrificed the mosquitoes, extracted total RNA, and synthesized cDNA for qRT-PCR. We performed qRT-PCR using four pools of mosquitoes injected with GFP dsRNA (*n* = 4) and *nkaβ2* dsRNA (*n* = 4) to assay for *nkaβ2* expression after RNAi treatment. The expression of *nkaβ2* mRNA was significantly reduced in *nkaβ2* dsRNA-injected mosquitoes relative to GFP dsRNA-injected mosquitoes (unpaired t-test: t = 3.0433, df = 6, *p* = 0.0227, Cohen’s *d* = 2.152618) (Figure 1b).

### 3.4. Na/K ATPase Subunit Beta Knockdown Does/Does Not Cause Significantly Increased Mortality Relative to Control

We injected two groups of female *Ae. aegypti* mosquitoes with either anti-GFP dsRNA (*n* = 67) or anti-*nkaβ2* dsRNA (*n* = 73). After 24 h, 83.6% of *nkaβ2*-injected mosquitoes were alive compared to 91% of GFP-injected mosquitoes (Figure 1c). After 48 h, 78.1% of *nkaβ2*-injected mosquitoes survived while 89.60% of GFP-injected mosquitoes remained alive (Figure 1c). After 72 h no more *nkaβ2*-injected mosquitoes had died (78.1% survival), and 85.1% of GFP-injected mosquitoes had still survived (Figure 1c). Overall, *nkaβ2* injection caused increased mortality at all time points measured PI but these differences were not statistically significant.

### 3.5. Na/K ATPase Subunit Beta Knockdown Has No Significant Effect on Ovary or Follicle Size

We injected two groups of female *Ae. aegypti* with either *nkaβ2* dsRNA or GFP dsRNA, and three days PI, we offered them a blood meal. We sampled injected mosquitoes at the following times post-blood meal (PBM): unfed, 24 h PBM, 48 h PBM, and 72 h PBM (Figure 2a). After blood feeding, mean NKAβ2 ovary lengths demonstrated a similar trend in growth to that of the GFP injected mosquitoes (Figure 2b). Follicle lengths between nkab2 dsRNA-injected and GFP injected mosquitoes were not significantly different at any time point measured (Figure 2c). 

### 3.6. Na/K ATPase Subunit Beta Knockdown Significantly Reduces Egg Numbers but Not Hatch Rates

We injected two groups of female *Ae. aegypti* with either *nkaβ2* dsRNA (*n* = 23) or GFP dsRNA (*n* = 25). Three days PI, we blood fed each group of mosquitoes and placed individual blood-fed mosquitoes into egg laying chambers. A total of 96 h PBM, we collected egg paper with clutches of eggs from each chamber. Egg counts from clutches laid by *nkaβ2* dsRNA-injected mosquitoes were significantly lower than egg counts from GFP dsRNA-injected mosquitoes (Mann–Whitney U: U = 171, df = 46, *p* = 0.0154, Cohen’s *d* = 0.576826) (Figure 2d). We observed a decrease in hatch rate of eggs laid by *nkaβ2* dsRNA-injected mosquitoes relative to eggs laid by GFP dsRNA-injected mosquitoes, but this difference was not statistically significant (Mann–Whitney U: U = 202, df = 46, *p* = 0.0781, Cohen’s *d* = 0.491782) (Figure 2e).

## 4. Discussion

*Ae. aegypti* mosquitoes are prominent vectors of many insect-borne diseases such as dengue, yellow fever, Zika, and chikungunya [46,47]. Mosquito-borne diseases have detrimental effects on the human population, infecting hundreds of millions of people every year [48]. Disease transmission is influenced by the need for female mosquitoes to acquire nutrient rich blood from hosts for the development of follicles within their ovaries. As more of the world becomes habitable for aedine mosquitoes due to accidental human transport and climate change [49], and growing numbers of mosquito populations evolve resistance to many currently used insecticides [50,51,52,53,54,55,56,57,58], novel means of reducing mosquito populations are necessary.

Nutrient-sensing pathways are promising targets for novel insect control treatments, as inadequate absorption of nutrients can cause adverse effects on female fecundity. After blood feeding, the insect fat body is responsible for uptake, metabolism, storage, and distribution of blood meal-derived nutrients during vitellogenesis [59]. Cationic amino acid transporters (CATs) are a vital component of sensing the digested amino acids from the blood meal and transporting positively charged amino acids into fat body cells [35,36]. We previously used a yeast two-hybrid assay to identify putative binding partners of the cationic amino acid transporter, CAT1 and identified NKAβ2 as a potential interacting partner [38]. In this study, we focused our efforts on understanding how *nkaβ2* RNAi knockdown affects female mosquito fecundity and have reported the effects of *nkaβ2* knockdown on survival rate, follicle size, ovary size, clutch sizes, and hatch rates.

Ion gradients established by NKA are vital for the survival of any organism, contributing to the reestablishment of the resting membrane potential [60], maintenance of cellular tonicity [61], and powering transport of other solutes across the cell membrane [62]. Disruption of ion transport by NKA in rabbit cells may also lead to secondary transport inhibition by improper ion gradient control [63]. Additionally, a lack of proper ion transport function leads to imbalanced ion concentrations causing cell death in *D. melanogaster* neural tissues [64]. Thus, we propose that knockdown of *nkaβ2* contributes to the decreased folding and localization of the catalytic NKAα subunit to the plasma membrane leading to the inhibition of neural functions and possibly contributing to the non-significant reduction in survival rates observed in *nkaβ2* knockdowns relative to controls (Figure 1c). Future studies analyzing the expression and localization of the NKAα subunit in response to *nkaβ2* knockdown will be important for determining how NKAβ2 may affect ion transport in neural and vitellogenic tissues.

We propose two hypotheses to explain the effects of *nkaβ2* knockdown on female fecundity. Our first hypothesis is that the effects of *nkaβ2* knockdown are due to reduced follicle cell patency in the ovaries of blood fed female *Ae. aegypti*. Induction of follicle cell patency during vitellogenesis is a well-documented mechanism of yolk protein passage to oocytes in many insects [65,66]. NKA pumps have been proposed to play a role in this process, as altered ion gradients in follicle cells, coupled with alterations in NKA subunit interactions at cell–cell junctions, can lead to shrinkage of follicle cells, producing intercellular gaps for yolk proteins to flow through [67,68]. Ouabain treatment has been demonstrated to decrease follicle patency, lending evidence to support the role of NKA pumps in this process [69]. Additional support for the role of NKA pump activity in follicle cell patency has been provided by studies demonstrating that signaling by hormones associated with vitellogenesis alter NKA pump activity in ovaries [68]. Interestingly, evidence of follicle cell patency in *Ae. aegypti* is mixed, with reports both reporting and refuting the opening of wider intracellular channels during vitellogenesis [47,70,71,72]. Whether follicle cell channels do indeed expand, evidence indicates that follicle cells seem to retract from the oocyte, leaving a wider perioocytic space for yolk nutrients to collect prior to uptake into the developing oocyte [70]. Decreased space for vitellogenin flux and storage for uptake by oocytes due to decreased NKA localization to the plasma membrane of follicle cells could be a possible mechanism by which the knockdown of *nkaβ2* leads to significantly decreased clutch sizes compared to the control group in our study. Knockdown studies using ouabain as a positive control for loss of NKA activity coupled with confocal and electron microscopy using fluorescent or electron-dense labels adapted from previously established protocols [70,73] would allow for elucidation of any potential role for *nkaβ2* in follicle cell patency. Additionally, detection of *nkaβ2* expression in ovaries after knockdown will provide crucial for determining if any effects observed in these experiments may be directly correlated with loss of *nkaβ2* expression in ovaries.

Our second proposed hypothesis for the role of *nkaβ2* knockdown in reduced fecundity is that reduction in cation transport across membranes of fat body cells alters the charge gradient of these cell membranes, decreasing transport efficiency of charged amino acids, which in turn reduces yolk precursor protein (YPP) synthesis. As described by [74], positively charged amino acids induce YPP synthesis through activation of the mTOR pathway, placing CATs in a central role during vitellogenesis [34,75]. Proteins that may interact with these CATs may then also be important for proper transport of amino acids, and activation of YPP synthesis in the fat body. While *nkaβ2* was expressed at highest levels in ovaries of *Ae. aegypti* as measured by qRT-PCR (Figure 1a), it was not significantly higher than in fat body, meaning that the effects of *nkaβ2* knockdown may be due to alterations in fat body NKA activity. Influx of cationic amino acids must alter the local membrane charge gradient, making it progressively more difficult to maintain the necessary amino acid flux without some way of maintaining the electrochemical gradient for these amino acids. Synthesis of amino acyl-tRNAs can maintain the chemical gradient for these amino acids, but the potential interaction between CAT1 and NKAβ2 that we previously identified [38] may be useful for stabilizing the local membrane charge gradient, thereby maintaining efficient intake of cationic amino acids. The knockdown of *nkaβ2* could possibly lead to insufficient synthesis of YPP’s, causing smaller clutch sizes as female mosquitoes must use the smaller amount of YPP’s to produce a smaller set of nutrient-rich oocytes. Research in *D. melanogaster* has demonstrated that nutrient deprivation leads to reduced fecundity [76]. When nutrient-deprived flies were fed a diet supplemented with essential amino acids, fecundity was increased compared to flies which were deprived of essential amino acids [76]. Additionally, RNAi knockdown of two other CATs, CAT2 and CAT3 significantly reduced mTOR signaling activity in *Ae. aegypti* after blood feeding [34]. As all three cationic amino acids (arginine, histidine, and lysine) are essential amino acids, and CAT1 is a histidine-specific transporter, these studies showing that essential amino acid depravation reduces fecundity in *D. melanogaster*, and CAT loss of function alters vitellogenesis provide support for our second hypothesis. Our lab has previously reported a method for culturing individual mosquito abdominal fat body organs to experimentally isolate the fat body from other mosquito organs [77]. Using this fat body culture system, radiolabeled amino acid uptake and YPP gene and protein expression studies in *nkaβ2* knockdown, or CRISPR/Cas9-mediated knockout mosquitoes using ouabain as a positive control for NKA loss of function can be performed to elucidate how reduced NKA function may affect YPP synthesis in *Ae. aegypti* mosquitoes. Additionally, ouabain treatment will affect the overall signaling and ion transport behavior of the NKAα subunit, so comparisons between ouabain-treated fat bodies and *nkaβ2* loss-of-function fat bodies will provide valuable insight into how *nkaβ2* loss-of-function affects NKAα activity, either by altering signaling, ion transport, or both.

In addition to the experiments described above to test each of our two stated hypotheses, metabolomic and proteomic studies of ovaries and fat body tissues at different time points post-blood meal in *nkaβ2* or GFP control dsRNA knockdown mosquitoes will also help to determine whether *nkaβ2* knockdown is affecting nutrient processing in the fat body or nutrient uptake in the ovaries. Dissections of treated mosquitoes after egg laying to look for any developed, but unlaid eggs in the ovaries may also provide insight into whether reduced egg counts are due to improper yolk deposition and egg development, or improper egg deposition. These experiments combined with microscopic analysis of follicle cell patency, and molecular analysis of amino acid transport in the fat body will be important for fully understanding how *nkaβ2* knockdown produces reduced egg numbers in *Ae. aegypti*. Along with the proposed metabolic and anatomical studies, knockdowns in mosquitoes at different time points post-eclosion may also provide valuable insight on what effect loss of *nkaβ2* function at different points before and after sexual maturity may have on the fecundity of these mosquitoes, and therefore how NKA activity at different times during sexual maturation may be important for female fecundity.

As the *nkaβ2* isoform we analyzed is orthologous to *D. melanogaster nrv3* (Appendix A), it is possible that *nkaβ2* may localize to similar tissues and serve similar functions as *nrv3*. *Drosophila nrv3* expression has been detected in a variety of cell types in different body regions including the central nervous system, Johnson’s Organs, chordotonal organs, and enteroendocrine cells [78,79,80]. Several of these regions, particularly the chordotonal organs which involve mechanosensation [81], and enteroendocrine cells that sense changes in midgut content and release chemical signals in response [78] have functions that could be analogous to important early post-blood meal processes of sensing abdominal stretch and midgut contents after blood feeding. Thus, it is possible that *nkaβ2* knockdown may influence *Ae. aegypti* fecundity by altering signaling of abdominal stretch receptors or enteroendocrine cells, thereby disrupting downstream nutrient and vitellogenic signaling, leading to decreased fecundity.

## 5. Conclusions

NKAβ2 plays a role in female *Ae. aegypti* fecundity as evidenced by smaller clutch sizes produced in *nkaβ2* knockdown mosquitoes relative to GFP knockdown controls. NKAβ2 expression patterns provide evidence for the development of multiple hypotheses to describe how *nkaβ2* knockdown affects *Ae. aegypti* egg numbers. Additional analysis needs to be performed to confirm the mechanistic pathways by which *nkaβ2* knockdown affects clutch sizes laid by female *Ae. aegypti*. However, the present research demonstrates an important role for the NKA pump in reproductive fitness of a medically important disease vector and provides a potential novel target for the design of new insect control treatments. 

## Figures and Tables

**Figure 1 insects-14-00050-f001:**
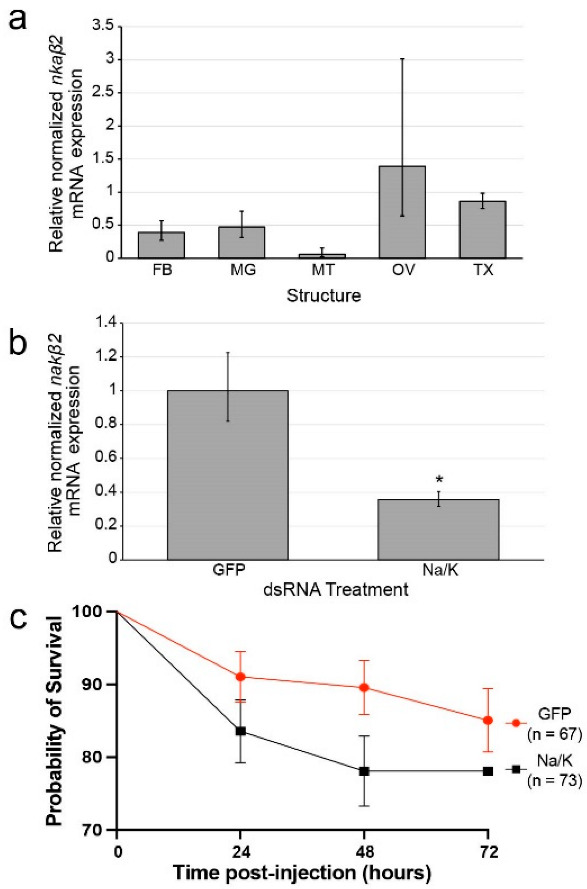
Profiling of *nkaβ2* expression and RNAi knockdown effect. (**a**). qRT-PCR reveals elevated expression of *nkaβ2* mRNA in ovaries relative to other tissues. While mean *nkaβ2* expression was highest in ovaries, expression was not significantly different between any tissue types. FB: fat body; MG: midgut; MT: Malpighian tubules; OV: ovary; TX: thorax. Statistical analysis was performed using a one-way ANOVA. (**b**). RNAi knockdown of *nkaβ2* (Na/K) results in significantly reduced *nkaβ2* mRNA expression in *nkaβ2* dsRNA-injected mosquitoes relative to GFP dsRNA-injected control mosquitoes 3 days PI. Significant differences between injected groups were determined using an unpaired t-test. Significant differences are marked with an asterisk. (**c**). RNAi knockdown of *nkaβ2* expression caused reduction in survival of female *Ae. aegypti* relative to GFP dsRNA-injected female *Ae. aegypti* through 72 h PI. Significance between survival curves was determined using a Mantel–Cox log-rank test.

**Figure 2 insects-14-00050-f002:**
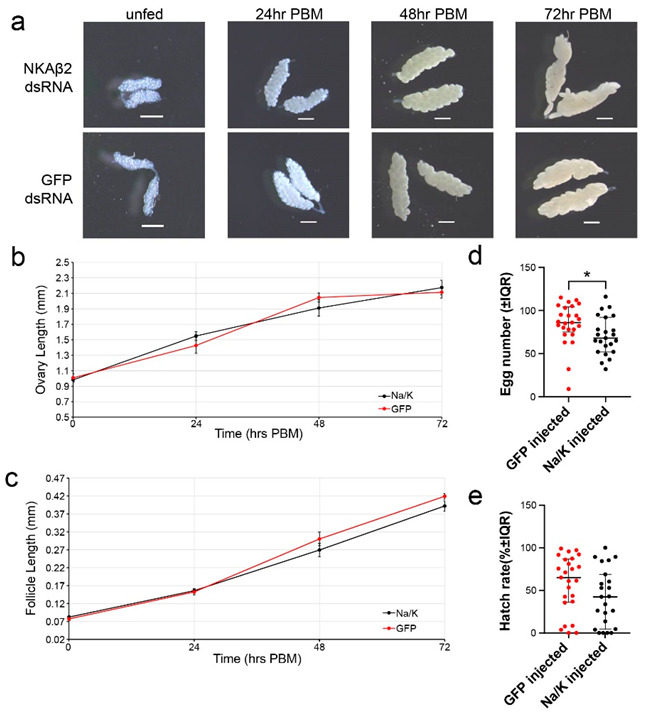
Effects of *nkaβ2* knockdown on vitellogenesis and fecundity. (**a**). Representative ovaries imaged from mosquitoes injected with either *nkaβ2* dsRNA or GFP dsRNA at stated times PBM. Scale bars in each image are 500 µm. (**b**,**c**). Average ovary lengths (**b**) and average follicle lengths (**c**) measured from pairs of ovaries dissected from 10 *nkaβ2* dsRNA-injected or 10 GFP dsRNA-injected mosquitoes at the following time points: unfed (0 h PBM), 24 h PBM, 48 h PBM, and 72 h PBM. Data points are represented as the average ovary lengths of 10 mosquitoes at each time point ± SEM (**b**) or the average of five individual follicle lengths measured from 10 mosquitoes at each time point ± SEM (**c**). Paired t-tests were used to determine significant differences in either average ovary lengths (**b**) or average follicle lengths (**c**) at each time point. (**d**,**e**). Egg numbers (**d**) and hatch rates (**e**) from *nkaβ2* dsRNA-injected (*n* = 23) and GFP dsRNA-injected (*n* = 25) female mosquitoes. Lines and whiskers represent median egg number or percent hatch rates ± interquartile range (IQR). Red dots and black squares represent data from individual mosquitoes. Statistical significance between median egg numbers (**d**) and hatch rates (**e**) was determined using Mann–Whitney U tests. Significant differences are marked with an asterisk.

**Table 1 insects-14-00050-t001:** qRT-PCR primers.

Primer	Sequence	T_m_ (°C)
*sodium/potassium ATPase subunit beta 2 (nkaβ2)* forward	TCCCACTGAGGAGCAGAAATACC	60
*nkaβ2* reverse	TGCTGCGGGCAAACTCTACC	
*ribosomal protein s7 (rps7)* forward *	TCAGTGTACAAGAAGCTGACCGGA	60
*rps7* reverse *	TTCCGCGCGCGCTCACTTATTAGATT	

* Internal reference gene.

**Table 2 insects-14-00050-t002:** dsRNA synthesis primers.

Primer	Sequence	Size (bp)
NKAβ2 forward	**TAATACGACTCACTATAGGGAGA**AATCGACTTCCTTCCTTTGGGG	618
NKAβ2 reverse	**TAATACGACTCACTATAGGGAGA**TTCTTCTTGGTGGTATGGCTCC	
GFP forward	**TAATACGACTCACTATAGGG**CGATGCCACCT	518
GFP reverse	**TAATACGACTCACTATAGGG**CGGACTGGGTG	

T7 promoter regions used for dsRNA synthesis are marked in bold. GFP dsRNA primers were previously reported [44].

## Data Availability

The data presented in this study are available in the main text of the article and in Appendix A.

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
