# Peer review of "Knockdown of the Sodium/Potassium ATPase Subunit Beta 2 Reduces Egg Production in the Dengue Vector, Aedes aegypti"

_insects, 2023, doi:10.3390/insects14010050_

Round 1

Reviewer 1 Report

Interesting study.

I have the following questions: 

1.    The authors chose to follow up on the survival of the mosquitos after the nkaβ2 knockdown only up to 72hrs. Did they look at the effect of the KD beyond the 72h time point as mentioned. If not, why?

2.    Why was GFP used a negative control?

3.    In fig1a and b, what are the nkaβ2 mRNA levels relative to? The error bar in fig1 a for the ovaries is quite bit. However, what is the significance of the higher levels of nkaβ2 in the ovaries?

4.    In Fig b, which organ/s are the nkaβ2 levels shown? Is the KD global, or it varies in different organs to different levels?

5.    In fig2, authors looked at the size of the ovaries after KD, but did they confirm the levels of nkaβ2 knockdown in ovaries specifically to attribute the effect?

6.    What is the significance/relevance of the information that “nkaβ2 is an ortholog of Drosophila melanogaster nervana 3”? Is it known to cause loss of fecundity or any other function in D.melanogaster? Is this KD known to affect fecundity of any other mosquitos?

7.    The authors suggest the following: “Thus, we propose that knockdown of nkaβ2 contributes to the decreased folding and localization of the catalytic NKAα subunit to the plasma membrane leading to the inhibition of neural functions and possibly contributing to the decrease in survival rates observed in nkaβ2 knockdowns relative to controls (Figure 1c).”

8.    Did the authors look at the levels of NKAα subunit in the nkaβ2 KD? Is there a known direct effect or correlation on the function of this protein in the absence of nkaβ2?

9.    The second hypothesis discusses the role of nkaβ2 KD affecting amino acid transport. Is there a known association or study that suggests this? Would this KD be affecting a global or overall reduction in signaling and function of the cells, rather than only specifically affecting Vitellogenesis as suggested by the authors?

Clearly this study shows there is an observed loss in egg production in female mosquitos, that the KD of nkaβ2 has shown, however, it is not clear how this may be happening. Does it affect egg development or egg laying? Is there a way to differentiate the two?

The authors have considered the limitation of this and hence provide hypothesis as explanation, but it is not very clear.

Reviewer 2 Report

The authors investigated the possible role of sodium/potasium ATPase subunit beta in the reproduction of the Dengue Fever mosuqito Aedes aegypti with majorly RNAi and qPCR methods. The research design is fine. The conclusion is supported by the results. Supprisingly, it is highly expressed in the ovaries, yet its knockdown did not affect the development of the ovaries. Instead, it has something to do with the survival of the insect. One would wonder what kind of roles it might play and how it works. 

This mosquito is sexaully matured around 3 days post eclosion. The authors chose 5-7 days old insects for the knockdown assay and the observation of the reproduction-related phenotypes and the analysis 3 days afterwards. Any specific reason? Is it because this gene is highly expressed within the 3-day window? Then how is it expressed developmentally ever since the adult eclosion?

There are a few minors that the authors are suggested to correct. 

Line 82, through,

Line 96, in,

Line 214-215, dsRNA was injected into the thorax, not through the mouth. 

Round 2

Reviewer 1 Report

Please check consistency for Gene name format: A. aegypti should be italicized throughout. 
